# The Natural History of Integrons

**DOI:** 10.3390/microorganisms9112212

**Published:** 2021-10-25

**Authors:** Timothy M. Ghaly, Michael R. Gillings, Anahit Penesyan, Qin Qi, Vaheesan Rajabal, Sasha G. Tetu

**Affiliations:** 1Department of Biological Sciences, Macquarie University, Sydney, NSW 2109, Australia; timothy.ghaly@mq.edu.au (T.M.G.); anahit.penesyan@mq.edu.au (A.P.); qin.qi@mq.edu.au (Q.Q.); vaheesan.rajabal@mq.edu.au (V.R.); 2ARC Centre of Excellence in Synthetic Biology, Macquarie University, Sydney, NSW 2109, Australia; sasha.tetu@mq.edu.au; 3Department of Molecular Sciences, Macquarie University, Sydney, NSW 2109, Australia

**Keywords:** evolution, metagenome, antibiotic resistance, lateral gene transfer, Anthropocene, resistome

## Abstract

Integrons were first identified because of their central role in assembling and disseminating antibiotic resistance genes in commensal and pathogenic bacteria. However, these clinically relevant integrons represent only a small proportion of integron diversity. Integrons are now known to be ancient genetic elements that are hotspots for genomic diversity, helping to generate adaptive phenotypes. This perspective examines the diversity, functions, and activities of integrons within both natural and clinical environments. We show how the fundamental properties of integrons exquisitely pre-adapted them to respond to the selection pressures imposed by the human use of antimicrobial compounds. We then follow the extraordinary increase in abundance of one class of integrons (class 1) that has resulted from its acquisition by multiple mobile genetic elements, and subsequent colonisation of diverse bacterial species, and a wide range of animal hosts. Consequently, this class of integrons has become a significant pollutant in its own right, to the extent that it can now be detected in most ecosystems. As human activities continue to drive environmental instability, integrons will likely continue to play key roles in bacterial adaptation in both natural and clinical settings. Understanding the ecological and evolutionary dynamics of integrons can help us predict and shape these outcomes that have direct relevance to human and ecosystem health.

## 1. Introduction

Integrons were discovered during research into the genetic basis of antibiotic resistance in Gram-negative pathogens [1]. They were shown to be an unusual kind of genetic element that could capture exogenous genes by site-specific recombination and then express these genes from an internal promoter [2,3,4]. The fundamental properties of integrons were established by examination of these early exemplars, now known as clinical class 1 integrons [5].

The study of clinical class 1 integrons established the general properties of all integrons. Functional integrons carry a gene for an integron-integrase (*intI*), which is a tyrosine recombinase (Figure 1). This enzyme catalyses site-specific recombination between the recombination site (*attC*) of a gene cassette and the integron attachment site (*attI*). Insertion of multiple gene cassettes results in a tandem array of cassettes. Gene cassettes are mobile, non-replicating elements, which generally consist of an open reading frame and an *attC* site, and are circular when not integrated into a cassette array [6,7].

Although most of the general information regarding integron structure and function is based on studies involving clinical class 1 integrons, this class is atypical in a number of ways. Clinical class 1 integrons are associated with plasmids and transposons, while the majority of integrons are found on bacterial chromosomes [8]. The clinical variants mainly carry gene cassettes that contain antimicrobial resistance genes, and because these cassettes have been acquired from diverse sources, they have highly variable *attC* sites [9,10]. In contrast, sedentary chromosomal integrons carry gene cassettes that encode genes that are mostly of unknown function and tend to have *attC* sites that are conserved in sequence and a secondary structure [10,11,12,13,14]. Lastly, clinical class 1 integrons have relatively short gene cassette arrays, usually comprising one to six cassettes, which are expressed from a single promoter embedded in the *intI* sequence (P_c_). Sedentary chromosomal integrons can have hundreds of cassettes in an array, and these arrays can have internal, cassette-borne promoters [15,16].

Consequently, while the class 1 integrons (and to a lesser extent, other mobile and clinically important integrons, classes 2 and 3) have allowed critical insight into the structure and function of these elements, they are not entirely representative of integrons as a whole. When considering the biology, ecology, and evolution of integrons, it is therefore important to consider the broader diversity of integrons, looking beyond what is observed in clinical samples and settings.

## 2. Integrons: Beyond the Clinical Environment

### 2.1. Diversity and Distribution of Integrons

Integrons are ancient elements involved in generating genomic diversity and driving bacterial adaptation [6,7]. Activity of the integron integrase is induced by the bacterial SOS and stringent responses and is therefore triggered in response to both DNA damage and nutrient starvation [17,18,19]. Consequently, integrons are able to generate genetic novelty at precisely the moment when it is needed the most, that is, during periods of stress or environmental change. Importantly, genetic variation can be rapidly generated while maintaining genomic integrity due to the modular and independent nature of newly inserted gene cassettes.

Integrons have been found in every environment surveyed [20,21,22,23] and are carried by diverse bacterial taxa [8]. To date, integrons have been detected within several phyla: Acidobacteriota, Actinobacteriota, Bacteroidota, Campylobacterota, Chloroflexota, Chrysiogenetota, Cyanobacteria, Desulfobacterota, Firmicutes, Gemmatimonadota, Proteobacteria, Planctomycetota, Spirochaetota, and Verrucomicrobiota [8,24]. They are particularly prevalent in Proteobacteria, where complete integrons can be detected in 20% and 10% of Gamma- and Beta-proteobacterial genomes, respectively [8] (Betaproteobacteria has since been reclassified as Burkholderiales, an order within the class Gammaproteobacteria [24]).

Recent metagenome studies indicate that integrons may indeed be even more diverse than indicated by surveys of sequenced genomes. Analysis of environmental samples has revealed that there are thousands of different integron ‘classes’ (discerned based on IntI amino acid homology) [25]. Similarly, the gene cassette repertoire of these environmental samples indicates that these elements are abundant and carry a diverse set of genes. Richness estimates suggest that between 4000 and 18,000 unique gene cassettes occur in as little as 0.3 g of soil [20]. The distribution of gene cassettes in bacterial taxa is even broader than the distribution of *intI,* as they are often present in so-called CALIN elements (clusters of *attC*s lacking an integron-integrase). Cury et al. [8] have proposed that CALINs could be formed by several mechanisms, including (i) off-target insertion of gene cassettes mediated by an IntI located elsewhere in a genome; (ii) loss of a recognisable *intI*, either through mutation, insertion or deletion; or (iii) genomic rearrangements that separate a cluster of *attC*s from its cognate *intI*.

### 2.2. Environmental Clustering of Integrons

Interestingly, integrons that share homologous IntIs and *attC*s are often found in divergent bacterial lineages that inhabit similar environments [10]. This can be inferred from the phylogeny of IntI protein sequences, which cluster according to the environment of origin rather than bacterial phylogeny. IntIs encoded by marine and soil/freshwater bacteria form the two distinct clades [7,10,16,25]. The same is true for *attC*s. Based on their sequence and structure, *attC*s also cluster according to their environment [10]. They form three major clades (Figure 2): One is a distinctly marine clade, the second is a soil/freshwater clade, and a third clade that has been named ‘Xanthomonadales-like’, as it is almost exclusively (>99%) comprised of *attC*s from the Xanthomonadales order. Xanthomonadales are commonly found associated with plant roots and leaves [10], suggesting that this clade might represent plant-associated environments.

This apparent incongruence between the phylogeny of integrons (IntIs and *attC*s) and their bacterial hosts has a number of potential explanations. It could result from horizontal transfer of integron platforms between lineages, likely to occur more readily where donor and recipient are co-located in the same environment. The environmental clustering of integrons could also be the result of convergent evolution, driven by selection for the ability of local IntIs to recognise *attCs* from the same environment [10].

Regardless of the evolutionary mechanism, the environmental partitioning of integron components has important implications for understanding integron ecology and the exchange of gene cassettes. Horizontal transfer of mobile gene cassettes is likely to be more prevalent among bacteria that co-exist in the same environment. Successful integration of a foreign gene cassette relies on the ability of the endogenous IntI to recognise and bind to the folded bottom strand of the gene cassette’s *attC* site [27,28]. However, different IntIs have been shown to recognise and recombine a different range of folded *attC* hairpin structures. For example, the class 1 integron integrase (IntI1) can recognise a much broader range of *attC*s than the *Vibrio cholerae* IntIA [29]. Thus, it is likely that similar IntIs can efficiently recombine similar ranges of *attC* substrates. This suggests that the exchange of cassettes might occur across broad phylogenetic boundaries due to shared IntI and *attC* homology within common environments. Indeed, at least among *Vibrio* isolates, the same gene cassettes can be detected even in the most distantly related species of the genus [30]. A selective advantage would be gained by integron platforms that can successfully recognise and integrate diverse cassettes from their local environment. It is not known whether this process generates the observed environmental clustering of integron components, or simply maintains it.

### 2.3. Functional Diversity of Gene Cassettes

While gene cassettes that carry open reading frames exhibit incredible sequence diversity, the full functional diversity of these putative genes remains unknown, as approximately 70–80% of gene cassettes encode hypothetical or uncharacterised proteins [19,24,30]. However, the analysis of cassette-encoded genes for which functional predictions are available indicates they contribute to a wide range of cellular activities, with examples found to be affiliated with all broad-level functional categories. The most commonly reported functions conferred by cassette-encoded proteins include toxin–antitoxin systems, glutathione S-transferases, acetyltransferases, receptor-associated transporters, and phage-related functions, such as restriction, methylation, and CRISPR systems [9,12,14,20,31,32,33,34]. Toxin–antitoxin gene cassettes can counteract homologous systems found on plasmids and bacteriophages, potentially protecting their host from invading mobile elements [35,36]. Together, these functions, which are largely associated with detoxification and phage-resistance, suggest that many cassette-encoded proteins are involved in the defence against biotic and abiotic stressors. Since many such stressors activate IntI activity, integrons increase the chances of conferring these prevalent cassette functions, which act to alleviate the stress. In addition, 10% to 30% contain signal peptide domains for membrane association or cellular export [12,19,35], and the most common cellular location has been identified as part of the membrane [32]. These findings support the idea that gene cassettes might play an important role in facilitating bacterial interactions with their hosts and broader environments.

Interestingly, among closely related strains, the gene cassette content of integrons can vary considerably, and most gene cassettes are only ever detected in a single integron [11,12,29]. Moreover, cassette arrays are highly dynamic, and the degree to which they change likely reflects the level of variability in the prevailing environmental conditions. Integrons are thus hotspots of genomic diversity and might facilitate niche specialisation.

We speculate that integrons and their cassette arrays represent variable and dynamic loci that are useful not only in responding to stressors, but also in facilitating bacterial adaptation to a range of different lifestyles and environmental niches. In providing a reservoir of genes that can be rapidly changed and rearranged, integrons represent a valuable commodity to facilitate bacterial survival and success in any number of situations. For example, among *Xanthomonas* species, gene cassette arrays are highly similar within the same pathovar, yet exhibit no similarity between different pathovars [11]. Although it is unknown what functions these cassettes encode, the strong correspondence between cassette array content and pathovar status suggests that they might play a role in facilitating species-specific interactions between *Xanthomonas* and their plant hosts. Further, many integrases among *Xanthomonas* have become inactivated, preventing further IntI-mediated changes to their cassette arrays [11,36,37]. Consequently, their arrays have stabilised after their selection as pathovars of specific plant hosts [11,38,39].

## 3. Integrons in the Anthropocene

The functionality of integron activity predisposes these elements to flourish in the Anthropocene. Human impacts on the microbial world have likely been more extensive than we know, resulting from changes in the global climate, altered nutrient cycles, loss of plant and animal biodiversity, and widespread pollution [37,38]. Specifically, the widespread use and subsequent environmental pollution with antimicrobial compounds place strong selective pressures on bacteria to acquire antimicrobial resistance genes. Integrons provide a mechanism for rapid movement and sharing of different resistance genes in response to changing antimicrobial selection pressures [39]. This obviates the need to stably maintain a diverse set of energetically costly resistance genes. However, prior to the Anthropocene, most integrons probably lacked an important feature that has now become the hallmark of the global resistance crisis: Mobilisation.

### 3.1. Mobilisation and the Rise of Class 1 Integrons

Sedentary chromosomal integrons are the ancestral state of integrons [14]. There are currently five known classes of integrons that have become mobilised by their insertion into conjugative elements. Of these, class 1 integrons are by far the most clinically important. They vector almost all known resistance gene cassettes that collectively confer resistance to most classes of antibiotics [40]. All mobile class 1 integrons are believed to have originated from a single ancestor in a Burkholderiales (formerly Betaproteobacteria) chromosome in the early 20th century [5].

Evolutionary reconstructions suggest that this ancestral chromosomal class 1 integron, carrying the biocide resistance gene *qacE*, was captured by a *res*-hunting transposon of the Tn*402* family [5,41]. The integron-carrying Tn*402* transposon was subsequently inserted into a mercury-resistance Tn*21*-like transposon. The *res* hunting function of Tn*402* allowed this mosaic element to insert into the *res* site of diverse mobile elements, subsequently facilitating its spread into a diverse range of bacteria. The now mobilised element encoding biocide and mercury resistance, which were used heavily prior to the antibiotic era, provided a significant selective advantage to its bacterial host. Subsequent insertion of the sulphonamide resistance gene *sul1* deleted the terminus of *qacE* to form *qacEΔ1*-*sul1*, which is now part of the 3′ conserved segment of clinical class 1 integron variants. Possession of *sul1* would again provide a selective advantage as sulphonamides became the first commercially available antibiotics in the 1930s [42]. Since then, the derivatives of this ancestral element have dramatically increased in abundance and distribution, spreading into diverse bacterial taxa [43]. Screening historic collections of pathogenic *Escherichia coli* show that class 1 integrons were not detectable in *E. coli* isolated before the 1940s, after which their prevalence rose dramatically to 26% of *E. coli* isolated in the 2010s [44].

### 3.2. Success of Class 1 Integrons

In addition to its mobilisation, the remarkable success of class 1 integrons is attributable to several key traits. The first is IntI1′s ability to recognise diverse *attC* substrates, providing it access to a larger pool of gene cassettes from diverse phylogenetic sources. Gene cassettes associated with class 1 integrons exhibit considerable sequence and structural variation between their *attC* sites, suggesting that they have originated from diverse genomic backgrounds [9,10]. Indeed, the degree of variation among *attC*s of class 1 integron gene cassettes is as broad as the total variation displayed by cassettes from all sequenced bacterial genomes (Figure 2) [10]. This suggests that IntI1 could potentially incorporate any gene cassette originating from any sedentary chromosomal integron.

The second is the lack of species-specific factors needed for IntI1 to catalyse cassette insertion, as is needed for other integron integrases [45]. For example, the endogenous IntI of *V. cholerae* is not active in other bacterial species as it relies on multiple *V. cholerae*-specific host factors [45]. It is unknown whether the chromosomal ancestor of class 1 integrons required such host factors and subsequently overcame these constraints during its evolution, or if they were never required for IntI1 activity. Regardless, it has allowed class 1 integrons to spread into, and be active within, a wide range of bacterial hosts, which is aided by their association with diverse transposons and broad-host range plasmids [6]. We now know that class 1 integrons have spread into at least 104 bacterial species from 44 genera (Appendix A). We inferred this from a BLASTP alignment of IntI1 (WP_000845048) against the RefSeq protein database (using 98% amino acid identity and 70% query cover thresholds) and cross-referenced the hits with the list compiled by Domingues et al. [43]. Even this is likely to be an underestimate of their true prevalence and distribution, based only on sequenced organisms contained within the RefSeq database.

Finally, the success of class 1 integrons can be attributed to the significant advantages they provide to their bacterial hosts under antimicrobial selection. Collectively, they have acquired more than 130 different resistance genes [40], with more being continuously discovered [46,47,48,49]. Despite class 1 integrons being able to recognise and integrate a broad range of gene cassettes, there seems to be a preponderance of antimicrobial resistance genes out of the broader diversity of cassette functions that are associated with class 1 integrons [25]. This overrepresentation of resistance genes among class 1 integrons can best be explained by two factors: (1) Strong positive selection imposed by the human use of antimicrobials, and (2) their ability to confer resistance phenotypes in a wide range of hosts without needing to integrate into metabolic networks [50]. The ability of clinical class 1 integrons to acquire diverse arrangements of these resistance cassettes has aided their spectacular rise in abundance and distribution.

Class 1 integrons can now be detected on every continent, including Antarctica [25,43]. Such is their abundance that millions to billions of copies of class 1 integrons are now present in a single gram of faeces from humans and agricultural animals [51]. This suggests that up to 10^23^ copies of these elements are being shed into the environment every day. Each of these integrons is presumably still active, and therefore capable of continuing to acquire different arrangements of resistance genes and other adaptive determinants.

### 3.3. Ongoing Integron-Driven Evolution in the Anthropocene

Class 1 integrons generally have short arrays of one to six cassettes that are expressed from a single promoter. The strength of cassette expression decreases with distance from the promoter [3]. During SOS-inducing stress, *intI1* expression is upregulated, leading to the acquisition of novel gene cassettes and rearrangement of existing cassettes, such that incoming cassettes are inserted at the start of the array where their expression is maximised [52,53]. Lineages with first-position cassettes that confer significant advantages are therefore likely to be selected. Experimental evolution shows that IntI1 activity accelerates the evolution of antibiotic resistance phenotypes [53]. Specifically, exposure to gentamycin resulted in the induction of IntI1 activity, which in turn increased the expression of a gentamycin resistance cassette by causing its duplication and insertion into the first position of the array. This resulted in a 64-fold increase in resistance to gentamycin [53].

Such integron-mediated evolution is likely happening across the globe. Indeed, class 1 integrons found inserted in the same plasmid backbone present in different bacterial isolates can vary considerably in their resistance cassette profiles [39]. Such variation in cassette array content likely represents adaptative responses to different antimicrobial selection pressures at a local scale. However, the implications are global. The world’s antibiotic usage continues to increase [54], resulting in significant antibiotic pollution radiating from human populations and agricultural areas [55,56]. Antibiotic concentrations with biological significance can now readily be detected in many natural environments [57]. Class 1 integrons, given their mobility, abundance, and global distribution, can rapidly provide their bacterial hosts with resilience to variable antimicrobial stressors.

## 4. The Future of Integron Evolution

Integrons will likely continue to play a major role in bacterial evolution in both clinical and environmental settings. Understanding the ecological and evolutionary dynamics of integrons can help us predict and shape these outcomes.

### 4.1. Natural Environments

Humans are driving environmental instability at an unprecedented rate [37]. Consequently, humans are inadvertently applying biotic and abiotic stress on microbes inhabiting almost all biomes on the planet [58]. Bacteria that can rapidly respond to environmental perturbations, occupy novel ecological niches, or degrade xenobiotic pollutants are likely to have significant advantages. Integrons have the potential to mediate such adaptation in their bacterial hosts. As such, we might expect strains of bacteria that carry integrons to have a selective advantage over related strains without integrons as environmental conditions continue to become more changeable.

Already, gene cassettes encoding xenobiotic-degrading enzymes can be readily recovered from polluted sediments, [59,60]. Positive selection for such genes imposed by anthropogenic pollution will likely facilitate their spread into diverse bacteria, as has been the case with antibiotic resistance cassettes. Adaptation to novel niches might also be aided by integron gene cassettes. For example, in submarine gas-hydrate-bearing sediments, gene cassettes can confer niche-specific functions that are metabolically relevant to their environment [61]. Environmental instability might provide opportunities for integron-carrying bacteria to invade specific niches and generate ecotypes. With improved sequencing technologies and methods for recovering integron sequences [25], we can investigate their ecological and evolutionary roles in natural environments more deeply.

### 4.2. Clinical Environments

The search for and use of novel antibiotics will undoubtably result in the subsequent spread of novel genes that can confer resistance to them. Integrons, particularly those of class 1, are almost certain to play a role in their dissemination. Early detection of such genes, via integron-targeted genetic surveillance, may provide an opportunity to flag new resistance mechanisms and develop mitigation strategies before they become widespread threats to clinical antibiotic efficacy. Additionally, integrons may pose a threat to alternative infection-treatment strategies, particularly phage therapy. Re-emergence of the therapeutic use of bacteriophage has provided a promising solution to combat antibiotic-resistant infections [62]. However, the high prevalence of phage-resistance functions encoded by integron gene cassettes raises concerns about the long-term success of such an approach [9,19,25,30]. Considering and managing the potential of integrons to hinder phage therapy is a necessity if we are to ultimately benefit from its therapeutic use.

Besides resistance, integrons might also drive the dissemination of other clinically relevant functions, particularly pathogenicity and virulence genes [50]. As antibiotic resistance becomes increasingly common in clinical settings, acquired genes that can enhance host colonisation and immune evasion will provide pathogens with an advantage over other resistant strains. There is evidence of this already happening. For example, class 1 integrons have acquired novel virulence cassettes and subsequently spread into multiple *Acinetobacter* species [39,63]. Virulence genes that can act as single units and function within multiple bacterial species are likely candidates to be integron-borne. Such genes could confer a broad range of functions such as those involved in cell surface adhesion, toxin production, biofilm formation, stress response, or nutrient acquisition and metabolism [50].

### 4.3. Integrons as a Biotechnological Resource

Integrons and their gene cassettes might provide a rich resource for biotechnology. Their value stems from two promising applications. The first is manipulating integron activity for genome engineering. The second is discovering gene cassettes with novel biochemical functions with biotechnological relevance.

The former has already been exquisitely demonstrated [64]. Applications of recombination-based genetic engineering tools are limited by the requirement of high sequence specificity of recombination sites. To overcome this, Nivina et al. [64] designed synthetic *attC* sites in silico, and used these as highly efficient, sequence-independent recombination sites for genomic engineering. As *attC*s rely on their folding structure to act as recombination substrates, they exhibit very minimal sequence-level constraints. As such, they can be embedded into virtually any protein-encoding gene without disrupting its amino acid sequence. Harnessing integron activity thus provides a unique structure-specific DNA recombination system that can be extremely useful for synthetic biology.

Additionally, manipulating integron integrase activity can be used to optimise gene and regulatory arrangements of biochemical pathways [65]. As a proof-of-concept, Bikard et al. [65] used the genetic shuffling activity of IntI1 to generate novel arrangements of a tryptophan biosynthetic operon. Several of these resulted in an order of magnitude greater yield of tryptophan than the natural gene arrangement. Such manipulations can be used to optimise industrially important biochemical pathways.

The second promising avenue is yet to be fully realised. The huge diversity and overrepresentation of genetic novelty encoded by gene cassettes make them a highly suitable resource of hitherto untapped enzymatic functions [50]. Since cassettes largely function as single-gene/single-trait entities, many are likely to be functionally active in a broad range of microbial hosts. This makes them highly useful commodities for synthetic biology applications. Here, IntI1 activity can be used for the recovery of gene cassettes from cloned genomic or metagenomic libraries, as has been demonstrated by Rowe-Magnus [66]. We might also select for novel genes that encode functions of interest by imposing artificial selection pressures [67]. In particular, their suspected roles in biodegradation, niche specialisation, and cross-species interactions suggest that many will encode useful functions that humans can leverage. Such functions could have applications in bioremediation, the development of environmentally friendly industrial processes, and enhancing rhizosphere and phyllosphere function for sustainable agricultural practices.

## 5. Conclusions

Our understanding of integrons and their remarkable roles in bacterial genome evolution has come a long way since they were first discovered more than two decades ago. Early studies on plasmid-borne clinical class 1 integrons established the fundamental principles of integrons and their roles in disseminating antibiotic resistance genes. Over time, the functional and evolutionary significance of integrons beyond antibiotic resistance has become increasingly recognised. Notably, advances in next-generation sequencing technologies allowed vast numbers of integrons from diverse bacterial species to be uncovered through metagenomic and genomic sequencing. Most of these gene cassettes encode hitherto unknown functions and represent a vast gene pool within the bacterial pangenome. Integrons are now considered to be ancient and diverse hotspots of genome innovation that are widespread in the chromosomes of many environmental bacteria.

Anthropogenic pressures on natural and human-modified ecosystems are projected to increase in the future [68]. This means that integrons will most likely continue to play key roles in conferring novel traits that benefit their bacterial hosts in their respective ecological niches. To better predict evolutionary trajectories of integron-mediated adaptation requires deeper insights into many outstanding questions in the field, including molecular mechanisms that are thought to contribute to novel gene cassette formation. It is hoped that future research will also pave the way for harnessing integrons as a biotechnological resource that improves sustainable agriculture, environmental remediation, and clinical treatment outcomes.

## Figures and Tables

**Figure 1 microorganisms-09-02212-f001:**
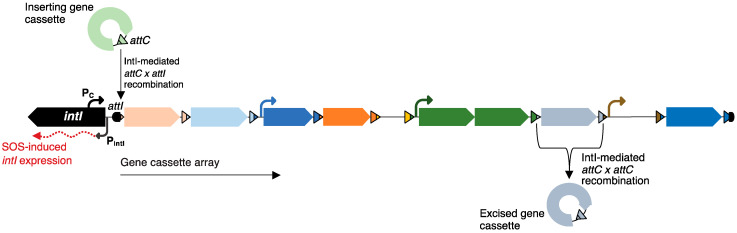
Structure and function of integrons. The typical structure of a chromosomal integron includes an integron integrase gene (*intI*), which encodes a tyrosine recombinase (IntI); an integron recombination site (*attI*); and a cassette array made up of 1–200+ sequential gene cassettes. All gene cassettes carry a cassette recombination site (*attC*) and generally consist of a single open reading frame (ORF), though cassettes that have two ORFs or no ORFs have also been observed. Some gene cassettes with two ORFs encode toxin–antitoxin systems. Cassettes closest to *attI* are strongly expressed from the cassette promoter (P_c_); however, some cassettes have their own promoters (bent arrows). Expression of *intI* (red, dotted arrow) is driven from the promoter (P_intI_) and is induced by the bacterial SOS response. IntI catalyses the insertion and excision of gene cassettes by mediating *attC* × *attI* and *attC* × *attC* recombination, respectively.

**Figure 2 microorganisms-09-02212-f002:**
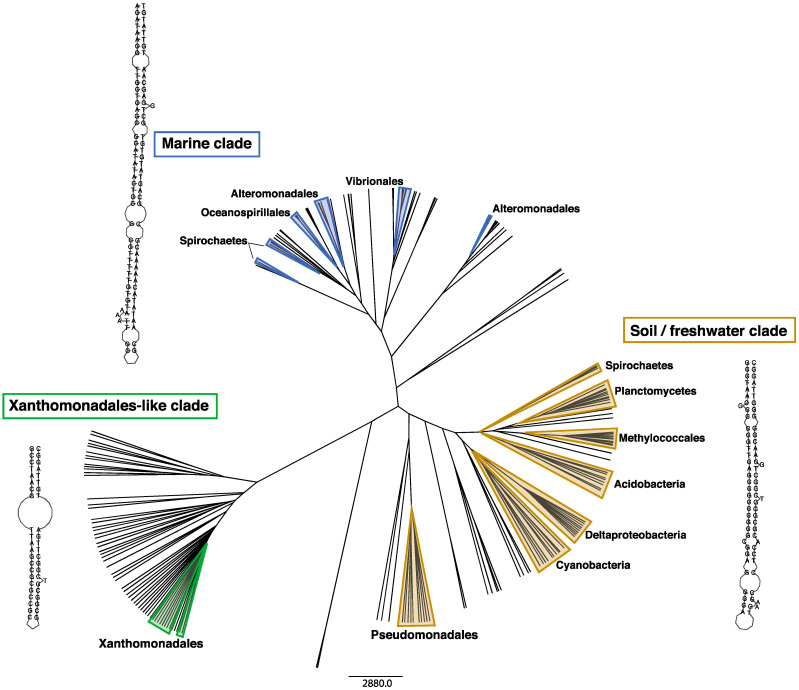
Environmental clustering of *attC* sequence and folding structures. The tips of the tree each represents an *attC* site. Tips outlined by shaded triangles represent *attC*s from chromosomal integrons. Unshaded tips represent *attC*s of antimicrobial resistance gene cassettes carried by class 1 integrons. Based on their sequence and structure, *attC*s cluster according to their environment, rather than host phylogeny, forming three major clades. Note that the sequence and structural diversity of *attC*s from class 1 integron gene cassettes spans that of all chromosomal *attC*s. The folded bottom strand of the best-fitting representative *attC*s from each clade are shown. These were determined by generating covariance models (CMs) built on the *attC*s from each clade, using previously described methods [10]. The *attC*s that best fit each model (based on CM bit scores) were selected as representatives and their structure predicted using RNAfold v2.4.16 from the ViennaRNA Package 2.0 [26].

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
