# Peer review of "The Natural History of Integrons"

_microorganisms, 2021, doi:10.3390/microorganisms9112212_

Round 1

Reviewer 1 Report

Nice review on the evolutionary history of integrons. I only have some minor comments mostly on mislocated refs.

L59-60. The first mention of this was in Rowe-magnus et al 2001 [43]

L92 integron”classes”: I think classes is a misleading terminology, I would substitute it to integron integrase genes. Classe suggest that they have different activities, which is not the case, it has been shown that integrases can insert cassettes in other classes of integrons (eg between 1 and 2).

Fig 2 is unreadable, of too poor quality.

L154-164: the refs are a bit too Australia centered, the Mazel group showed restriction enzymes and TA cassettes in 2001 and 2003 [refs 12 and 43], and showed that TA cassettes could counteract potentially those of plasmids in 2007 [PMID: 17259320] and 2013 [PMID: 23475970]. The fact that this TA cassettes can be plasmid resistance module, should be added L 158.

L168-169: Not true for V. cholerae, the many strains share more than 10%!

L355-358: It is worth mentioning that gene cassette capture setup have been developed and published such as in Rowe-Magnus PMID 19596808, as well as the synthetic integron setup for improving metabolite production published in Bikard et al PMID 20534632.

L448-454: refs 31 and 32 are identical to 33 and 34.

Author Response

Nice review on the evolutionary history of integrons. I only have some minor comments mostly on mislocated refs.
            >>>> We thank the Reviewer for their positive feedback.

L59-60. The first mention of this was in Rowe-magnus et al 2001 [43]
            >>>> This reference has been added as suggested

L92 integron”classes”: I think classes is a misleading terminology, I would substitute it to integron integrase genes. Classe suggest that they have different activities, which is not the case, it has been shown that integrases can insert cassettes in other classes of integrons (eg between 1 and 2).
            >>>> Although, we completely agree with Reviewer 1, the term “class” is still used extensively in the literature, particularly for mobile integrons. We also would like to use consistent terminology, as parts of this manuscript specifically discuss “class” 1 integrons. However, to address Reviewer 1’s concerns, we have corrected the sentence to: “Analysis of environmental samples has revealed that there are thousands of different integron ‘classes’ (discerned based on IntI amino acid homology)” [Lines 91-93].

Fig 2 is unreadable, of too poor quality.
            >>> We thank Reviewer 1 for pointing this out. The resolution of Fig 2 must have been altered during the submission reformatting process. The original high-res figure has been re-inserted here.

L154-164: the refs are a bit too Australia centered, the Mazel group showed restriction enzymes and TA cassettes in 2001 and 2003 [refs 12 and 43], and showed that TA cassettes could counteract potentially those of plasmids in 2007 [PMID: 17259320] and 2013 [PMID: 23475970]. The fact that this TA cassettes can be plasmid resistance module, should be added L 158.
            >>> We have now included the suggested references. In particular, we would like to thank the Reviewer for raising the point that TA cassettes can potentially confer resistance to invading mobile elements. This has now been included in the manuscript [Lines 158-160]

L168-169: Not true for V. cholerae, the many strains share more than 10%!
            >>>> This sentence has now been removed from the manuscript.

L355-358: It is worth mentioning that gene cassette capture setup have been developed and published such as in Rowe-Magnus PMID 19596808, as well as the synthetic integron setup for improving metabolite production published in Bikard et al PMID 20534632.
            >>>> As suggested, we have now mentioned both of these studies [Lines: 352-357 & 363-365].

L448-454: refs 31 and 32 are identical to 33 and 34.
            >>> We thank Reviewer 1 for picking up on this. We have now corrected this reference duplication.

Reviewer 2 Report

The work by Ghaly et al. provides a beautiful review about the evolutionary and ecological dynamics of integrons. The authors present a well-organized and comprehensively described overview on this topic, which I had the pleasure to read and learn more about.

In the pdf I received, Figure 2 shows up with very low resolution and it's hard to read the labels. I would improve the figure quality. Other than that, I have nothing else to add. I would like to congratulate the authors for this very interesting piece of science!

Author Response

We thank Reviewer 2 for their positive comments. Regarding Figure 2, we thank the Reviewer for pointing this out. The resolution of Fig 2 must have been altered during the submission reformatting process. The original high-res figure has been re-inserted here.